# Prognostic Role of sST2 in Acute Heart Failure and COVID-19 Infection—A Narrative Review on Pathophysiology and Clinical Prospective

**DOI:** 10.3390/ijms23158230

**Published:** 2022-07-26

**Authors:** Luca Marino, Antonio Concistrè, Marianna Suppa, Gioacchino Galardo, Antonello Rosa, Giuliano Bertazzoni, Francesco Pugliese, Claudio Letizia, Luigi Petramala

**Affiliations:** 1Department of Mechanical and Aerospace Engineering, “Sapienza” University of Rome, 00161 Rome, Italy; 2Emergency Medicine Unit, Department of Emergency-Acceptance, Critical Areas and Trauma, Policlinico “Umberto I”, 00161 Rome, Italy; marianna.suppa@uniroma1.it (M.S.); giuliano.bertazzoni@uniroma1.it (G.G.); antonello.rosa@uniroma1.it (A.R.); goacchino.galardo@uniroma1.it (G.B.); luigi.petramala@uniroma1.it (L.P.); 3Department of Clinical, Internal, Anesthesiological and Cardiovascular Sciences, Sapienza University of Rome, 00161 Rome, Italy; antonio.concistre@gmail.com (A.C.); claudio.letizia@uniroma1.it (C.L.); 4Department of General Surgery and Surgical Specialties “Paride Stefanini”, “Sapienza” University of Rome, 00161 Rome, Italy; f.pugliese@uniroma1.it; 5Department of Translational and Precision Medicine, “Sapienza” University of Rome, 00161 Rome, Italy

**Keywords:** biomarkers, sST2, acute heart failure, COVID-19, risk stratification

## Abstract

The importance of cardiovascular biomarkers in clinical practice increased dramatically in the last years, and the interest extends from the diagnosis purpose to prognostic applications and response to specific treatment. Acute heart failure, ischemic heart failure, and COVID-19 infection represent different clinical settings that are challenging in terms of the proper prognostic establishment. The aim of the present review is to establish the useful role of sST2, the soluble form of the interleukin-1 receptor superfamily (ST2), physiologically involved in the signaling of interleukin-33 (IL-33)-ST2 axis, in the clinical setting of acute heart failure (HF), ischemic heart disease, and SARS-CoV-2 acute infection. Molecular mechanisms associated with the IL33/ST2 signaling pathways are discussed in view of the clinical usefulness of biomarkers to early diagnosis, evaluation therapy to response, and prediction of adverse outcomes in cardiovascular diseases.

## 1. Introduction

The search for useful biomarkers that can be adopted in clinical practice are of paramount importance, particularly in the acute phase of many diseases, in order to improve diagnostic precision as well as prognostic significance. Biomarkers increase their role in the management of many diseases; in the diagnostic procedure, the values of many biomarkers represent a crucial step in the diagnostic algorithm and in the decision tree. In the acute phase of the presentation of the pathology the adoption of specific and sensitive biomarkers can reduce time to act, optimize treatment, and improve knowledge of prognosis. In many cases biomarkers provide important contributions to the prognostic risk stratification and can be adopted to track the disease state in a follow-up analysis after discharge from the hospital.

In the last 20 years, among the several biomarkers considered, attention on sST2 has significantly increased and it is now considered one of the most promising markers for the clinical management of many diseases.

As a member of the interleukin-1 receptor superfamily, the protein ST2 (alternative name for “interleukin-1 receptor-like 1”) presents with a transmembrane (ST2L) and soluble (sST2) isoforms. Since the discovery of interleukin-33 (IL-33) in 2005 [1] as the ST2L ligand, the molecule has been deeply considered in inflammatory conditions, fibroproliferative diseases, autoimmune diseases, trauma, sepsis, and most recently and significantly in pulmonary and cardiovascular diseases. The molecular pathophysiologic role of sST2 was initially associated with chronic conditions, and more recently, also in the acute phase of the diseases mentioned [2,3,4].

In this review, we focus on the possible role of the sST2 as a prognostic tool to stratify the outcome of patients admitted for acute heart failure alone or associated with COVID-19 infections.

## 2. The Molecular Setting

Localized on chromosome 2 (2q12), the ST2 gene encodes different isoforms through alternative promoter splicing. In particular, the transmembrane (ST2L) and the soluble (sST2) ones are particularly important in the inflammation and fibrosis processes [5].

### 2.1. IL-33/ST2L Axis Signaling

ST2L is expressed through several cells (basophiles, CD4 lymphocytes, eosinophils, macrophages, and keratinocytes) and its role is tightly linked to the release of its natural ligand, the interleukin-33 (IL-33). Nowadays it is usual to consider the IL-33/ST2L axis to the full comprehension of the inflammatory cytokines/chemokines production and the successive regulation of innate and adaptive immune systems to promote inflammatory responses. Historically, the ST2 (also called the IL-33 receptor, T1, DER4, and FIT-1) was described in advance of the detection of IL-33, with potential confusion in terminology. ST2 was originally cloned as an oncogene-induced gene from murine fibroblasts and later a second similar ST2 mRNA transcript was detected and predicted to code for a receptor, now identified as the transmembrane-bound ST2L receptor [6]. The first molecule identified was the secreted isoform and it is now designated an sST2 [7].

IL-33, identified by Schmitz et al. in 2005 [1], belongs to the Toll-like/IL-1-receptor superfamily and can be considered an alarmin which signals tissue damage. In fact, it is secreted by most cells in response to exposure to pathogens, injury-induced stress, or death by necrosis [8]. Its expression has been observed in many organs such as stomach, lung, spinal cord, brain, skin. The strongest expression was found in endothelial cells, epithelial cells, keratinocytes, fibroblasts, fibrocytes and smooth muscle cells [9]. The action of IL-33 is exerted through two mechanisms: as a nuclear factor that directly binds to chromatin in the nucleus and as a cytokine joining to ST2L.

In physiologic conditions, the intracellular localization of IL-33 is mainly in the nucleus, where it regulates gene expression in several ways; particularly as a direct action in suppressing the NF-kB-regulated genes that are necessary for pro-inflammatory signaling [10] but also as an epigenetic modulator via histone deacetylase-3 (HDAC3) [11].

In pathological conditions such as in the necrosis processes, IL-33 reveals its “alarmin” nature and it is released by the cell in the extracellular space. By binding to the ST2L receptor, IL-33 exerts its cellular functions through several signal pathways. In particular, the molecular function is obtained by the action of the heterodimeric ST2L/IL-1RAcP complex on a variety of immune cells, where the IL-1R accessory protein (IL-1RAcP) plays an essential role by enhancing the affinity of IL-33 for ST2L.

In more details, the binding of the IL-33 to ST2L allows the receptor to experience a conformational change, which enables the recruitment of IL-1RAcP [7]. Heterodimerization of the two transmembrane molecules conveys the two intracellular TIR domains together. Successive receptor adaptor proteins are recruited, and the signal is transmitted by the adaptor proteins myeloid differentiation factor-88 and IL-1R-Associated Kinase (MyD88-IRAK1), respectively, and the 4-TRAF6 signaling pathway with resulting degradation of the inhibitory protein IκB and subsequent activation of the NF-κB transcription factor [1]. Furthermore, MAP kinases, p38, JNK, and ERK are also activated with triggering of downstream transcription factors, such as AP-1, that enables transcription of cytokines and chemokines in a cell-type restricted manner.

Activation of this pathway in Th2 cells leads to production of Th2 cytokines (i.e., IL-4, IL-5, IL-13) [1], whereas in epithelial cells, ST2 activation by IL-33 results predominantly in chemokine activation [12].

During wound healing, extracellular IL-33 interacts with the ST2L receptor and the complex of IL-33/ST2L activates myeloid differentiation primary response 88 (MyD88) intracellular cascades that drive production of type 2 cytokines (such as IL-13) from polarized Th2 cells. Most recently, tissue fibrosis, mucosal healing, and wound repairment were found to be additional possible actions of IL-33 during inflammation processes.

### 2.2. IL-33/sST2 Molecular Role

The second variant of the ST2 gene expression is the circulating soluble ST2 (sST2), which is shorter than ST2L. It is identical in structure to the extracellular region of the long ST2L isoform except for nine additional amino acids at the C-terminus and it is obtained [3,13] through differential mRNA processing. It can be produced spontaneously in the lung, kidney, heart, and small intestine and after activation with IL-33 in mast cells or with anti-CD3/anti- CD28 in both CD4 and CD8 T cells. Its expression is largely inducible and ubiquitous in living cells. The role of sST2 was recently investigated as an IL-33 soluble receptor and blocker of effects in target cells [1]. Specifically, sST2 avidly binds to IL-33 and prevents its fastening to ST2L in the immune cell (mainly lymphocytes T). In this framework, sST2 can be thought of as a decoy receptor. As a consequence, it can inhibit the activation of Th2 cell response and the release of anti-inflammatory cytokines (IL-4, IL-5, IL-10, IL-13), polarizing the Th1 response, which results in the activation and release of inflammatory cytokines (TNF-α) and inflammation.

Figure 1 provides a sketch of the two different isoforms ST2d and sST2 with the possible biological actions.

## 3. The Clinical Setting

### 3.1. Acute Heart Failure

While pro-Brain Natriuretic Peptide (pro-BNP) is widely recognized as the gold standard marker for the diagnosis of acute heart failure (HF) in routine clinical practice, several studies have evaluated the use of different markers with potential prognostic value in HF patients; novel molecules such as sST2, emerge as potentially useful biomarkers, providing additional diagnostic and prognostic value with different and controversial findings in relation to the studied population and the end points evaluated. The American Heart Association/American College of Cardiology guidelines on the management of HF recommend the measurement of sST2, in addition to other fibrosis biomarkers, in patients with acute HF for a more appropriate stratification [14]; moreover, sST2, unlike NT-proBNP, is not influenced by age, body mass index (BMI), renal function, or the etiology of HF [15].

In a prospective single-center study, pro-BNP and sST2 showed high diagnostic power for HF (AUC 0.976 and 0.889, respectively), but sST2 revealed stronger power to predict fatal events (in-hospital and 1 month mortality rates), as well as other markers of negative prognosis such as the use of inotropes or high lactate levels [16]. The American heart failure cohort study (PRIDE study) [17] firstly analyzed the role of sST2 in defining acute HF in patients enrolled in emergency department with acute dyspnea, and NT-proBNP was significantly superior to sST2; however, this study suggested that sST2 was more important as a prognostic tool, regarding mortality, due to HF. Regarding this prognostic aspect, sST2 and NT-proBNP showed additive significance; in fact, the increase in both markers showed the highest mortality rate after 1 year and 4 years (about 40%). This study suggested a threshold of sST2 ≥ 35 ng/mL as a predictor for poor prognosis and risk of death [17].

In a large Asian population, Yamamoto M et al. evaluated the additional clinical value of sST2, Pentraxin 3, Galectin-3, and high-sensitive cardiac troponin (hs-TnT) beyond BNP for risk stratification, showing that sST2 was associated with significant outcomes (all-cause, cardiovascular mortality, and HF hospitalization) in patients with acute decompensated HF solely in subjects with preserved ejection fraction [18]. On the other hand, in a large study conducted by Edmin M. et al. [19] in a population with a prominent reduced ejection fraction (more than 80% with reduced left ventricular ejection fraction (LVEF) <40%) in a multivariate model (including age, sex, BMI, ischemic etiology, LVEF, NYHA classification, glomerular filtration rate, medical therapy, NT-proBNP, and hs-TnT), the risk of all-cause death, cardiovascular death, and HF hospitalization were increased by 26%, 25%, and 30%, respectively, per each doubling of sST2 [19]. The same results were obtained in the prospective study by Zhang R. et al.; higher levels of sST2 showed a significantly higher rate of adverse outcomes (all-cause mortality or heart transplantation), strictly correlated to left ventricular ejection fraction [11]. In addition, several multicenter cohort studies (i.e., MOCA study and ASCEND-HF study) showed that sST2 concentration had high ability in predicting CV mortality at short term (30 days) as well as long term (1 year) [20,21].

In the prospective cohort study, Wang Z. et al. evaluated 331 patients affected by acute HF according to sST2 levels, and followed progression for almost 2 years; patients with higher behaviors of sST2 had a higher left ventricular mass index, lower left ventricular ejection fraction, higher NYHA score and higher NT-proBNP levels; moreover this group showed the worst primary outcome (cardiovascular mortality) in all patients with acute AF [22].

The Oulu Project Elucidating Risk of Atherosclerosis (OPERA) Survey collected data during a long-term follow-up (21 years), evaluating several data about morbidity and mortality. The main determinant of sST2 levels during the follow-up was smoking habit, and diabetes in the multivariate model male sex; levels of alanine aminotransferase (ALAT), high-density lipoprotein (HDL) cholesterol, and high-sensitive C-Reactive Protein (hsCRP) were associated with elevated sST2 levels. Moreover, sST2 levels were higher among subjects suffering from cardiovascular disease, cancer, mild cognitive decline, and diabetes. ST2 was found as an independent predictor of total mortality, evaluated with several covariates (age, sex, diabetes, smoking, transaminases, HDL-cholesterol, and hsCRP) [23].

Conversely, in the Controlled Rosuvastatin Multinational Trial in Heart Failure (CORONA) cohort, sST2 was significantly associated with secondary endpoints (worsening HF and hospitalization due to worsening HF) in older patients with chronic HF, whereas sST2 did not show prognostic significance regarding primary endpoint (CV death, non-fatal myocardial infarction, or stroke) pro-BNP, and C-reactive protein [24].

Interestingly, a significant aspect resides in the variation in sST2 levels during the treatment of HF; in fact, more recent studies claim that continuous measurements greatly increase the amount of available prognostic information, and the use of sST2 as a monitoring tool [2]. The Prospective Randomized Amlodipine Survival Evaluation 2 (PRAISE-2) study conducted on severe chronic heart failure (NYHA class III to IV) suggested that the variation in sST2 during hospitalization, more than baseline sST2 value, is a significant predictor of end-point, beyond well-known predictive significance of baseline BNP and baseline precursor peptide of atrial natriuretic peptide (ProANP) [2]. In a meta-analysis conducted on a large population that was followed for more than 1 year, Aimo A et al. showed that both values of sST2 obtained at admission and at discharge from hospitalization had significant prognostic value regarding all-cause and CV deaths, but sST2 at discharge was more predictive of HF re-hospitalization during the follow-up (HR higher than 2.5 times) [25].

### 3.2. Ischemic Heart Failure

Beyond mortality and prognosis related to HF, Zhang Q et al. evaluated the usefulness and ability of sST2 to suggest the degree of coronary artery stenosis and to predict the development of adverse events 1 year after acute coronary syndrome (ACS). In this regard, the authors evaluated sST2 at discharge after hospitalization due to non-ST segment elevation acute coronary syndromes (NSTE-ACS) treated with percutaneous coronary intervention (PCI); the subgroup with higher levels of sST2 (>34.2 ng/mL) was associated with higher Gensini scores and multivessel disease, resulting as a marker predictive for the no-reflow phenomenon (AUC 0.662, sensitivity 66.7% and specificity 65.2%, OR 3.802). Moreover, sST2 > 34.2 ng/mL had a significant role to predict 1-year prognosis, with a significantly higher rate of major cardiovascular and cerebrovascular events (MACCE) (HR 10.22) [26].

An interesting registry-based study (registry SWEDEHEART) conducted on STEMI and NSTEMI patients followed for 6.6 years, showed that beyond other markers, sST2 was strongly useful to discriminate STEMI from NSTEMI patients, but lacking predictive ability to identify death or major adverse cardiovascular events; as a limit of this study, the lack of significance was probably due to the fact that a single dosage was taken at the time of admission, and not repeated during the long-term follow-up [27].

Recently Park S. et al. evaluated the predictive ability of sST2 to detect left ventricular (LV) remodeling after ACS. Patients with reduced LV ejection fraction (EF < 50%), underwent PCI for ACS (unstable angina, non-ST-elevation myocardial infarction, and ST-elevation myocardial infarction), evaluated echocardiographically at baseline and at a 3-month follow-up. During the follow-up, a significant correlation between sST2 and changes in LV end-diastolic/systolic volume index (r = 0.649 and r = 0.618, respectively) was found but not in the changes in LVEF; moreover, the reduction in sST2 was more predominant in patients without adverse remodeling. Non-correlation was found in relation to ACS types. A multivariable analysis showed that sST2 variation was the most important determinant of LV remodeling following the revascularization of ACS [28].

### 3.3. Heart Disease and SARS-CoV-2 Infection

Recent studies showed that high levels of sST2 were detected during COVID-19 infection in patients without cardiovascular comorbidities positively correlated with CRP behaviors, suggesting an emerging role of sST2 as an inflammation marker during SARS-CoV-2 infection with stronger diagnostic value (ROC area 0.9896) [29], and associated with a worse outcome. Several studies found that higher levels of sST2 levels (above 58.9 ng/mL) are associated with significant AUC (0.776) for ICU admission, mechanical ventilation, or in-hospital death [30].

Unlike mild cases of COVID-19 infection characterized by an increased number of regulatory T (Treg) cells and a scavenger-like phenotype of alveolar macrophages (suggesting a strong immune response leading to an adequate viral clearance and tissue integrity restoration) [31], in severe forms of COVID-19, especially with bilateral pneumonia, injury of alveolar epithelial cells promotes an accelerated release of IL-33, that may paradoxically upregulate sST2 and its own receptor on Treg cells, with inhibitory effects [31], immune intolerance and an increased secretion of type-2 proinflammatory cytokines [32], and decreased counts of CD4+ and CD8+ T cells [29].

In SARS-CoV2 patients, Luft T et al. found that in subgroup with severe disease (treated with mechanical ventilation and/or death) higher endothelial stress index (EASIX), useful predictor of endothelial activation, and life-threatening complications, was correlated to behaviors of several endothelial markers, as well as sST2, thrombomodulin, angiopoietin-2, CXCL8, CXCL9, and interleukin-18 [33].

There are some promising preliminary results from trials using anti-ST2 antibodies such as Astegolimab [31], that can represent a potential therapeutic target for controlling excessive inflammation-related injuries in pulmonary or cardiovascular systems in patients with moderate to severe forms of COVID-19 [31].

Moreover, a high prevalence of cardiovascular diseases among SARS-CoV-2-infected patients and vice versa has been reported, as well as significant risk of severe forms of COVID-19 among patients with preexisting HF [5,34], also demonstrated by the fact that cardiac biomarkers present an high prognostic value in the assessment of COVID-19 severity, predicting mortality rates.

Evaluation of sST2 behaviors can be useful in order to detect those patients at high risk of developing myocardial injury due to ACE2-related cytotoxic effects, determining onset of HF, either de novo or an exacerbation of a chronic HF [35]. Likewise, higher levels of sST2 can underlie those patients with a history of ischemic HF at high risk of plaque rupture during systemic inflammation due to COVID-19 infection [36]. Evaluation of sST2 behaviors can also add advice regarding pharmacological treatment. Gaggin et al. reported that use of a higher beta-blocker dosage was more effective in subjects with increased values of sST2 [37].

Moreover, it is well-known that during SARS-CoV2 infection, acute respiratory distress syndrome (ARDS) is associated with right ventricle (RV) failure [38] and development of pulmonary hypertension (PH), which can benefit from the cardiac resynchronization therapy (CRT). Beaudoin et al. demonstrated a significant sST2 increase at the six-month follow-up in patients with persistent PH and progressive PH and patients without the development of PH, [39], suggesting a predictive role of sST2 also in PH.

The limit of the sST2 analysis in COVID-19 infection is provided by the few findings reported and most importantly, non-uniform studies in literature. The disease is still in a fast evolution phase, but future investigations must be designed to address the specific role for diagnostic and prognostic purposes, in particular with prospective follow-up studies.

Table 1 reports a scheme of the characteristics of the analyzed studies, particularly with the relationship to the prognostic role of sST2.

## 4. Conclusions

Although several molecules have been investigated in the last decades for diagnostic, therapeutic, and prognostic purposes, their use can have a diversified use in relation to clinical use and the different characteristics of the molecule. The IL-33/ST2 axis was initially evaluated in relation to the ability to influence neoplastic growth, and later in relation to cardiovascular diseases, inflammatory conditions, fibroproliferative diseases, autoimmune diseases, and systemic infections.

Focusing attention on cardiovascular diseases, international guidelines and studies conducted on large populations suggest that, beyond its limited use on diagnosis of HF respect other biomarkers (i.e., pro-BNP, hs-CRP, and hs-TnT), the evaluation of sST2 is extremely useful in correlation to clinical data, in particular on the severity of the underlying cardiomyopathy, left ventricle hypertrophy, cardiac remodeling, ejection fraction, NYHA score, and finally to obtain prognostic information, especially regarding medium- to long-term mortality.

Moreover, sST2 is correlated to the severity of ischemic heart disease; in fact, it strongly suggests poor prognosis with higher involvement of stenotic coronaries, high rate of coronary artery re-stenosis after percutaneous coronary intervention (PCI), worse myocardial remodeling post-PCI, and higher rate of major cardiovascular and cerebrovascular events during long-term follow-up.

Finally, beyond bi-directional correlations between cardiovascular disease and SARS-CoV-2 infection, sST2 can be useful in evaluation and risk stratification in patients with SARS-CoV-2 infection; in fact, this biomarker is correlated with worse evolution of the infection, evaluated as ICU admission, mechanical ventilation needed, or in-hospital death.

The role of the sST2 in COVID-19 is actually promising but still not completely clear. Further observational and prospective studies are needed to evaluate the diagnostic and the risk stratification function for the disease and its comorbidities. In particular, future analyses need to be designed to clarify some aspects, such as timing of sampling (i.e., admission or discharge) or evaluation of its changes after specific treatment. The prognostic role, particularly promising, must be investigated through follow-up studies on medium- (months) and long-terms (years). In this case, serial samples together with clinical data and instrumental exams, can help to address the disease evolution and consequences.

## Figures and Tables

**Figure 1 ijms-23-08230-f001:**
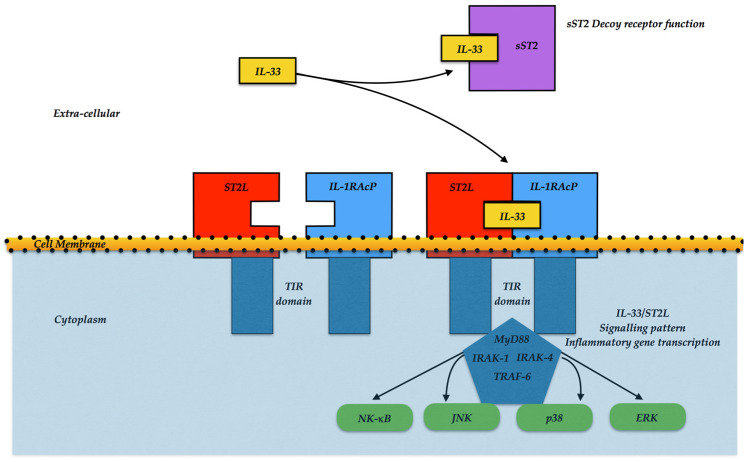
IL-33/ST2L signaling pattern. IL-33 can bind to the ST2/IL-1 receptor accessory protein (IL-1RAP) heterodimer, then enroll MyD88 to its intracellular domain. Alternatively, the IL-33 binds to the sST2 *decoy* receptor, impairing the further signal. MyD88 fastening involves IL-1R-associated kinase (IRAK) and TRAF6, leading to either the NF-κB, JNK, p38, or ERK activation, and promoting inflammatory cytokine expressions.

**Table 1 ijms-23-08230-t001:** Characteristics of the analyzed studies.

Study	Sample	Disease	Follow-Up	sST2 Cut-Off Value or X-times of the Mean Value	Major Findings
Miftode et al., 2021 [16]	120	AHF	1 month	60 ng/mL	Prognostic for fatal events. OR 3.3.
Jannuzzi et al., 2007 [17]	593	AHF	1 year	35 ng/mL	Prognostic for death. OR increases linearly with sST2 concentration.
Yamammoto et al., 2021 [18]	616	AHF	3 years	17 pg/mL	Prognostic of CV death and HF rehospitalization. OR 1.422 per unit increase in the natural logarithm of the sST2.
Edmin et al., 2018 [19]	4268	CHF	2.4 years	28 ng/mL	Prognostic for CV death and HF hospitalization.
Lassus et al., 2013 [20]	5306	AHF	30 days 1 year	76 ng/mL	Risk stratification for death.
Tang et al., 2016 [21]	858	AHF	6 months	71.2 ng/mL	Prognostic for increased death risk. OR 2.21.
Zhang et al., 2021 [22]	105	AHF	1 year	2122.65 ng/mL	Prognostic for HF re-admission or death. Correlation between lipoprotein-associated phospholipase and sST2.
Filali et al. [23]	600	MULTIPLE	21 years	23.7 ng/mL	Prognostic for total mortality. OR 9.9 per unit increase in the logarithm of the sST2.
Aimo et al., 2017 [25]	4835	AHF	13.5 months	2 X	Prognostic for all-cause (OR 2.06) and cardiovascular (OR 2.20) death, HF hospitalization (OR 1.54).
Zhang et al., 2021 [26]	205	NSTE_ ACS	1 year	34.2 ng/mL	Prognostic for MACE. OR 10.22.
Hjort et al., 2021 [27]	1082	NSTEMI-STEMI	6.6 years	4.6 ng/mL (STEMI) 4.2 ng/mL (NSTEMI)	Prognostic for all-cause mortality (OR 1.36), MACE (OR 1.32).
Park et al., 2021 [28]	95	ACS	3 months	32 ng/mL	Predictive for LV remodeling. OR 1.24.
Zheng et al., 2022 [29]	80	COVID-19	no	147 pg/mL	Serum sST2 associated positively to CRP and negatively to lymphocytes T (CD4^+^, CD8^+^). OR 5.87 per unit increase in the logarithm of the sST2.
Sanchez et al., 2021 [30]	152	COVID-19	no	58.9 ng/mL	Predictive for ICU admission or death
Luft et al., 2021 [33]	100	COVID-19		2X of endothelial activation and stress index	sST2 positively associated with endothelial activation and stress index. OR 3.4 for worse outcomes.

sST2: soluble ST2; AHF: acute heart failure; OR: odd ratio; LV: left ventricular; CPR: C reactive protein; ICU: intensive care unit; NSTEMI: non-ST elevation myocardial infarction; STEMI: ST elevation myocardial infarction; ACS: acute coronary syndrome; MACE: major cardiovascular and cerebrovascular events.

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
