# Peer review of "Prognostic Role of sST2 in Acute Heart Failure and COVID-19 Infection—A Narrative Review on Pathophysiology and Clinical Prospective"

_ijms, 2022, doi:10.3390/ijms23158230_

Round 1

Reviewer 1 Report

I think that authors made a very complete revision of the subject and the topic is highly relevant to clinical practice. I think that the review is publishable in his present form.

Author Response

Thank you for consideration and opinion.

Reviewer 2 Report

The search for the most informative, highly sensitive and specific biomarkers, as well as those with high prognostic value, still does not lose its relevance. Indeed, a successful search for such a marker would significantly improve the diagnostic algorithm, optimize treatment and monitor the patient's adherence to therapy. sST2 has demonstrated high diagnostic and prognostic value in heart failure in numerous studies. A significant percentage of large single and multicenter studies have shown greater predictive value in both the short and long term compared to pro-BNP, and the combined use of these markers increases predictive power. However, of course, there are many controversial and controversial points. A global problem affecting the whole world - the COVID-19 epidemic has opened the prospect of new research indicating the growing role of sST2 as an inflammatory marker during SARS-CoV-2 infection.

In this review, the authors have compiled key research findings on sST2 in HF, data from 2020-2021, and the latest research findings on its role in COVID-19. The sources of the literature used are dated mainly from 2020-2022. However, the data in this area are not so numerous. Of course, the problem “is young”, but I think it’s worth somehow reflecting this in the section on sST2 and COVID-19, perhaps there are some research limitations, etc. Also in the Conclusions section, a lot is written about sST2 and CVD, but, in my opinion, the main conclusions about the role of sST2 in COVID-19 are not fully reflected, what are the prospects for further research in this area, what exactly requires further study?

In general, the review is structured, fairly concise, written in good scientific language, without unnecessary thought. The literature used is up-to-date. The percentage of literature used in the last 5 years is 64%.

In conclusion, it is worth noting that minor comments are advisory in nature and the review “Prognostic role of sST2 in acute heart failure and COVID-19 infection. A narrative review on pathophysiology and clinical prospective.” may be published in the International Journal of Molecular Sciences.

Author Response

We thank Reviewer 2 for the important comments and suggestions. We added a comment displaying the limits and pitfalls of the research of sST2 in COVID-19.

In the Conclusions, we added further comments enhancing the limits of the sST2 role at the moment and the possible future studies characteristics, necessary to give the biomarker the correct role in the clinical scenario.

Reviewer 3 Report

In this review by Luca Marino et al, the authors summarized the prognostic roles of sST2 in different settings of heart failure as well as cardiac injury/infection. The manuscript is informative and provides new insights into the understanding of heart failure prognosis in clinics by using sST2 as a marker.

In the section 2 the molecular setting, I would suggest the authors to use subtitles to clarify the genome difference and their signaling pathways respectively.

The authors introduced ST2L and sST2 as isoforms of ST2. It would be better if the authors provide a diagram to show the difference.

The authors cited a series of clinical studies and summarized the diagnostic and prognostic roles of sST2. I would expect the authors to plot a chart to show the quantitative data of sST2 volume and the relationship to prognosis.

Author Response

We thank Reviewer 3 for the helpful suggestion that improve the readability of the manuscript. We introduced two subsections for a better comprehension of the section 2 (Molecular Setting).

Following the precious suggestion of Reviewer 3, we added a figure to clarify the IL-33/ST2L signaling pathway and the role of sST2. We are confident that this change will increase the readability.

We again thank Reviewer 3 for this opportunity. We have added a detailed table where the sST2 analysis has been summarized and the prognosis item has been enhanced for each contribution.